# PARP Inhibitors in the Treatment of Early Breast Cancer: The Step Beyond?

**DOI:** 10.3390/cancers12061378

**Published:** 2020-05-27

**Authors:** Anthony Gonçalves, Alexandre Bertucci, François Bertucci

**Affiliations:** 1Medical Oncology Department, Cancer Research Center of Marseille (CRCM), INSERM U1068, CNRS UMR7258, Institut Paoli-Calmettes, Aix-Marseille University, 13009 Marseille, France; bertuccia@ipc.unicancer.fr (A.B.); BERTUCCIF@ipc.unicancer.fr (F.B.); 2Predictive Oncology Laboratory, Cancer Research Center of Marseille (CRCM), Inserm U1068, CNRS UMR7258, Aix-Marseille University, 13009 Marseille, France

**Keywords:** BRCA, early breast cancer, PARP inhibitors

## Abstract

Exquisitely exploiting defects in homologous recombination process, poly(ADP-ribose) polymerase (PARP) inhibitors have recently emerged as a promising class of therapeutics in human epidermal growth factor receptor 2 (HER2)-negative advanced breast cancer with germline breast cancer 1 (BRCA1) or breast cancer 2 (BRCA2) mutations (gBRCA1/2m). In this setting, PARP inhibitors, either as single agent or in combination with platinum-based chemotherapy, significantly increased progression-free survival, as compared to conventional chemotherapy. Accordingly, further therapeutic advances are expected at an earlier stage of the disease. In the neoadjuvant setting, veliparib failed to increase the pathological complete response rate when added to a carboplatin-based regimen, in unselected triple-negative breast cancer patients. Similarly, when administered before anthracycline-cyclophosphamide, the neoadjuvant olaparib-paclitaxel combination was not superior to carboplatin–paclitaxel, in patients with HER2-negative breast cancer and BRCA1/2 mutation, or homologous recombination defect. Yet, neoadjuvant talazoparib, administered as a single-agent in patients with HER2-negative breast cancer and germline BRCA1/2 mutation, achieved an impressive pathological complete response rate of nearly 50%. In the adjuvant setting, the results from the OlympiA phase III study, evaluating adjuvant olaparib in HER2-negative early breast cancer and germline BRCA1/2 mutations, are eagerly awaited. Ongoing trials should clarify whether PARP inhibitors might improve outcome when administered in the adjuvant or neoadjuvant setting in early breast cancer patients with BRCA1/2 mutation or homologous recombination defect.

## 1. Introduction

In less than 5% of cases, breast cancers (BC) are associated with germline mutations in breast cancer 1 (BRCA1) or breast cancer 2 (BRCA2) (gBRCA1/2m) genes. These mutations are involved in nearly 30% of hereditary BC and predict a high probability of the disease during a lifetime [1,2], reaching more than 70% in the most recent studies [3]. In gBRCA1/2m carriers, emerging BC usually lack the expression or function of BRCA1/2 proteins, with severe consequences on the DNA repair process, most notably of double-strand breaks (DSB), using the homologous recombination (HR) process [4]. This is supposed to result in genomic instability, which might explain the excess of some cancer risks (not only breast but also ovarian, prostate, or pancreatic cancer). Importantly, while BC in gBRCA1m carriers are predominantly basal-like, luminal subtypes are most frequently found in gBRCA2m, as is the case in the sporadic cancer population [5,6].

Through their role in the base excision repair process, poly(ADP-ribose) polymerase(PARP) family members, notably PARP1, are key players in the repair of DNA single-strand breaks (SSBs) [7]. In presence of PARP inhibitors (PARPi), unrepaired SSBs accumulate and might convert into DSBs at the replication fork [7,8]. In HR-defective cells, such as gBRCA1/2m, DSBs repair relies upon alternative and error-prone mechanisms, and thus PARPi might ultimately lead to cell death, according to the so-called “synthetic lethality” [9]. Of note, in vitro cytotoxicity of PARPi highly correlates with their ability to trap PARP–DNA complex at SSBs sites [10]. Table 1 shows the name and chemical structure of major PARPi in clinical development. The synthetic lethality concept has been successfully tested in the clinic during the last 10 years, with various PARPi demonstrating target inhibition in peripheral and tumor tissues, as well as a favorable toxicity profile and a promising clinical efficacy as a single-agent, in various advanced cancers associated with gBRCA1/2m, including BC [11]. Accordingly, several PARPi have been approved in the management of advanced ovarian cancers in various settings, mostly as maintenance following platinum-based chemotherapy, in patients carrying mutations of BRCA1/2, either germline or somatic, but also in wild-type BRCA1/2 (platinum sensitivity serving as a functional surrogate of BRCAness) [9]. In BC, olaparib and talazoparib have been approved as a single-agent in patients with human epidermal growth factor receptor 2 (HER2)-negative advanced BC and gBRCA1/2m. Both agents were evaluated in randomized phase III trials (OlympiAD and EMBRACA, respectively) enrolling anthracyclines and taxanes pre-treated, and endocrine-resistant if ER+, patients [12,13]. Patients received either PARPi or standard chemotherapy according to the physician’s choice (vinorelbine, capecitabine, eribuline, or gemcitabine). In both studies, progression-free survival (PFS, the primary endpoint) was significantly increased, as well as objective response and clinical benefit rates, and quality of life was improved in the PARPi arm when compared with the chemotherapy arm. Significant but manageable side effects included hematological (notably anemia) and digestive toxicities as well as asthenia. Of note, no overall survival (OS) benefit was reported in the OlympiAD trial [14] and EMBRACA trial, as recently reported at the AACR 2020 meeting. More recently, the addition of low-dose veliparib (VELI) to carboplatin–paclitaxel was tested versus placebo in HER2-negative, gBRCA1/2m, advanced BC (BROCADE 3 trial). In case of chemotherapy discontinuation for toxicity, veliparib was pursued at standard dosage, as maintenance [15]. Veliparib increased PFS—most of the effect being observed during the maintenance phase—while moderately affecting the toxicity profile of chemotherapy. Again, no impact on OS has yet been reported.

Based on these promising results in the metastatic setting, PARPi are expected to play a significant role in the early disease. In this paper, we review the recent clinical data as well as ongoing trials evaluating this issue and discuss the perspectives of use of this compound class in this setting.

## 2. PARPi Added to Chemotherapy in the Neoadjuvant Setting

Preclinical data indicate a potential synergism between PARPi and chemotherapy, notably with DNA-damaging agents [16]. However, the combination of chemotherapy and PARPi has been limited by frequent hematological toxicity. In terms of PARP-trapping activity, a feature supposed to correlate most with PARPi cytotoxic activity, VELI has been considered as the weakest PARPi under clinical assessment, which could be viewed as an advantage, potentially rendering its combination with chemotherapy more feasible. It was the first PARPi assessed in the neoadjuvant setting in the carboplatin-VELI arm of the ISP-Y 2 trial. The I-SPY 2 TRIAL (Investigation of Serial Studies to Predict Your Therapeutic Response Through Imaging and Molecular AnaLysis 2, I-SPY 2) was an adaptive randomized phase II study aiming to evaluate various innovative therapeutic compounds in the neoadjuvant setting, in specific subsets of BC defined by combining the expression of HER2 and hormone receptors, with the Mammaprint signature [17,18]. Patients with specific signatures were then randomized to receive either a backbone control chemotherapy regimen including 12 weekly paclitaxel, followed by 4 adriamycin-cyclophosphamide (AC) cycles every 2/3 weeks (P-AC), or the same regimen in which the experimental drug was added to the paclitaxel sequence. The primary outcome was the pathological complete response (pCR) rate and the study provided a Bayesian estimation of pCR in each signature evaluated, as well as the probability of success of the experimental arm in a future 300-patient randomized phase III trial. In this trial, the carboplatin-VELI arm enrolled 72 patients with HER2-negative BC, including both hormone receptor-positive (HR+) and hormone receptor negative (HR−, i.e., triple-negative BC, TNBC), who received oral VELI 50 mg twice a day (BID) during 12 weeks and 4 cycles of 3-weekly carboplatin AUC 6 during the paclitaxel sequence (VELI-CARBO) [19] (Figure 1). The pCR was estimated to 33% (95% Bayesian probability interval PI 23–43%), versus 22% (95% PI 10–35%) in 44 patients treated with the control arm. The benefit of VELI-CARBO seemed to be restricted to the TNBC patients (51% for VELI-CARBO vs. 26% in the control arm in TNBC; 14% vs. 19% in HR+/HER2−); the probability of success of a large phase III randomized trial was estimated to 88% in TNBC patients (vs. 53% in the overall HER2-negative population, and 8% in HR+/HER2− patients). Of note, there were more gBRCA1/2m patients in the VELI-CARBO arm (17%) compared to the control arm (5%). Hematological toxicities, including grade 3 or more neutropenia (71% vs. 2%), thrombocytopenia (21% vs. 0%), and anemia (28% vs. 0%), were significantly increased in the VELI-CARBO arm. Yet, with regards to either efficacy or tolerance, it was not possible to distinguish the relative impacts of only the VELI, only the CARBO, or a combination of both. Of note, ancillary studies from the I-SPY2 samples revealed that a DNA repair deficiency signature, including presence of a gBRCA1/2m status, was associated with a higher response to the VELI-CARBO [20,21].

Based on these results, a randomized phase III trial (BrighTNess) was subsequently conducted in 634 TNBC patients receiving neoadjuvant chemotherapy evaluating VELI-CARBO vs. placebo-CARBO or double placebo, in combination with 12 weekly paclitaxel followed by 4 cycles of AC [22]. VELI and CARBO were administered similarly as in the I-SPY2. The pCR rate was higher in the CARBO-containing regimen (53% in VELI-CARBO and 58% in the placebo-CARBO vs. 31% in the control arm), but was not different between the VELI-CARBO and the placebo-CARBO arms. Of note, there was no clear impact of the gBRCA1/2m status on the pCR rate. Within the patients with gBRCA1/2m, the pCR rate was 57% in the VELI-CARBO/paclitaxel arm, 50% in the placebo-CARBO/paclitaxel arm, and 41% in the control paclitaxel arm. The difference was not significant, but the trial was not powered to detect statistically differences between these subgroups. No survival data were available. Hematological and digestive toxicities were more frequent in CARBO-treated patients, and CARBO increased delays or dosage reduction of paclitaxel, but only slightly affected the overall paclitaxel delivery, while the addition of VELI increased anemia and thrombocytopenia but did not increase the non-hematological toxicities, and had a minimal effect on paclitaxel exposure. The risk of febrile neutropenia during subsequent AC was increased in CARBO-treated patients. Interestingly, a recent secondary analysis of this study investigated the surgical outcomes and found that in a small subgroup of patients considered as not eligible for breast-conserving surgery at diagnosis (*n* = 141), a similar percentage of patients became eligible for conservative surgery in the double-placebo versus placebo-CARBO arms (44.1% in both groups), but there was a non-significant numerical trend in favor of the VELI-CARBO arm (61%, *p* = 0.11). In patients who were initial candidates for breast-conserving surgery, the rate of progression of treatment was significantly lower in the CARBO-containing regimen (4.8% in the VELI-CARBO, 5% in the placebo-CARBO, and 12.8% in the double placebo arm, *p* = 0.01) [23].

Even though the biological rationale to combine PARPi and anti-microtubule agents remains limited, the GeparOLA study examined association of the olaparib (OLA) with paclitaxel in HER2-negative patients with homologous DNA repair deficiency [24]. This German Breast Group’s study was a non-comparative randomized phase II trial investigating the pCR rate after neoadjuvant chemotherapy, with low-dose OLA (100 mg BID, continuously 12 weeks) administered concomitantly with weekly paclitaxel ×12, followed by EC ×4 in 69 patients with gBRCA1/2m, or somatic BRCA1/2 mutation, or a high homologous recombination deficiency (HRD) score (as defined by genome-wide copy number analysis on tumor DNA) and HER2-negative BC. A control group including 37 patients was treated with weekly carboplatin–paclitaxel ×12, followed by EC ×4. OLA-treated patients had lower serious adverse events and treatment discontinuation than with carboplatin. With a pCR rate of 55.1%, the OLA-containing arm failed to reach its primary endpoint (to exclude a pCR rate lower than 55%) and had a similar activity as a carboplatin-based arm (48.6%). However, promising activity for the OLA-arm vs. the CARBO-arm was noted in HR+ (52.6% vs. 20%), younger patients (76.2% vs. 45.5%), and HRD score high/wtBRCA (51.7% vs. 37.5%).

In the 3-stage randomized phase II/III PARTNER trial, OLA was associated with a combination of 3-weekly CARBO + weekly paclitaxel for 4 cycles before standard anthracycline-based chemotherapy. TNBC with basal phenotype or any gBRCA1/2m with T1–T4 N0–2 M0 BC were eligible. The first two stages included a 1.1.1 randomization between the control arm and the 2 OLA-based schedules (OLA 150 mg BID Day 2 to 10 or Day 3 to 14) and were to evaluate the safety and to select the best schedule. A total of 159 patients were included during the stages 1 and 2 and tolerance data were recently reported—most frequent grade 3 or more events were neutropenia (19%), anemia (15%), and thrombocytopenia (5%), but febrile neutropenia and hemorrhage were uncommon. Extra-hematological toxicities included fatigue and diarrhea. Toxicities were regarded by authors as not exceeding historical frequencies with conventional chemotherapy. The efficacy stage 3 of PARTNER study is ongoing with the OLA D3-14 schedule [25].

Taken together, the few available results from the neoadjuvant studies evaluating PARPi, in combination with chemotherapy have not yet demonstrated a significant advantage over standard or carboplatin-based regimen, either in unselected TNBC or in HRD-positive/BRCA1/2 mutated early BC patients. Hematological toxicities render it difficult to combine these drugs with cytotoxics, notably carboplatin, and make it necessary to use low-dose schedules. Additional data are still needed and other trials testing a combination of PARPi with neoadjuvant chemotherapy are ongoing.

## 3. PARPi as Single-Agent in the Neoadjuvant Setting

There are only limited data on antitumor activity of PARPi when administered as single-agent, before surgery in the early setting. The idea is that PARPi monotherapy might optimize PARP inhibition while minimizing toxicity. A pre-operative phase I study with pharmacodynamics and pharmacokinetics endpoints was conducted in 60 patients receiving five dose levels of OLA BID during 4–5 days before surgery. The study revealed a lower plasma exposure than anticipated but significant drug concentration in tumors and PARP inhibition in peripheral blood cells, as well as in tumor tissues [26].

More recently, a pilot trial was performed at the MD Anderson Cancer Center to evaluate a short course of oral talazoparib (TALA) 1 mg once a day (OD) over 2 months, before starting the standard neoadjuvant chemotherapy in gBRCA1/2m patients with HER2-negative early BC. TALA, considered to be the most powerful PARPi in terms of “PARP-trapping” capacity, a feature that correlates with the cytotoxic effect in vitro, might be the best candidate for single-agent treatment in this setting. The primary objective was to evaluate the feasibility of enrolling 20 patients in two years, but the study was prematurely stopped because of rapid accrual (13 patients in 8 months) and significant evidences of anti-tumor activity—all patients had a clinical ultrasound assessment-based response (median decrease in tumor volume of 88%), 10 out of 13 had a residual cancer burden (RCB)-0-I, 3 had RCB-II, and none had RCB-III after TALA single-agent followed by conventional chemotherapy. No grade 4 toxicity was observed and there were 7 grade 3 toxicities including 3 neutropenia, 2 anemia, 1 thrombocytopenia, and 1 leukopenia, only 1 leading to dose reduction. All patients received and completed post-TALA conventional chemotherapy [27].

In view of these promising results, a subsequent expansion cohort was started in which gBRCA1/2m, HER2-negative, stage I-III BC patients received neoadjuvant TALA as single-agent during 4 to 6 months, without any chemotherapy, in order to evaluate the pCR rate without chemotherapy and tolerance [28]. A total of 20 patients were enrolled, including 16 with gBRCA1m status, 15 TNBC, 12 stage II and 3 stage III patients (Figure 2). Nineteen patients completed 6 months of treatment and 10 of them had pCR (RCB-0: 53%) while two additional patients had RCB-I. The rate of RCB 0–I was 63% overall, 57% in TNBC, and 80% in HR+, 53% in gBRCA1m, and 100% in gBRCA2m. Of note, pCR was observed in tumor types with lower probability of response to conventional treatment, such as lobular cancer, inflammatory breast cancer, and metaplastic chondrosarcomatous carcinoma. Twelve patients received post-operative chemotherapy (mostly TNBC patients). Tolerance was consistent with previous clinical experiences with TALA, including anemia (8 patients received erythrocyte transfusion during treatment), mild nausea, and fatigue. Of note, there was no apparent impact on menstruation during the TALA treatment. Based on 33 patients, these data might be viewed as preliminary but they represent the first results of pathological complete response after a single PARPi monotherapy in breast cancer. They compare favorably with other reported results in gBRCA1/2m cohorts of early BC patients receiving neoadjuvant cytotoxic chemotherapy, with either platinum-based or platinum-free regimens. A confirmatory multicentric, phase II study is ongoing (NCT03499353), testing the same TALA regimen in the same neoadjuvant setting. Importantly, various patient reported-outcomes will be collected, which might help evaluating the impact of TALA on quality of life, an important endpoint if PARP monotherapy is challenging conventional chemotherapy in this setting. Expected reports of metastasis-free survival would be interesting. Unfortunately, the lack of a control arm in this study will not allow a direct comparison with neoadjuvant chemotherapy in terms of either efficacy or tolerance. Another ongoing study is evaluating single-agent niraparib in gBRCA1/2m-operable BC patients, with a change in tumor volume evaluated by breast magnetic resonance imaging (MRI) as the primary endpoint (NCT03329937). Table 2 recapitulates the neoadjuvant studies testing PARPi (either with chemotherapy or as a single-agent) in early BC, with available results.

## 4. PARPi in the Adjuvant Setting

The presence of a residual disease after neoadjuvant chemotherapy is a strong predictive factor for survival in TNBC, identifying patients with the highest risk of recurrence, which might represent a favorable setting to evaluate the efficacy of experimental post-operative treatments. A US collaborative group conducted a comparative study enrolling 128 BC patients with TNBC or gBRCA1/2m (22 of 101 tested), which has received neoadjuvant chemotherapy and had residual lymph node-positive or mammary residual invasive disease > 2 cm. Patients were randomized between adjuvant cisplatin 75 mg/m^2^ D1 every 3 weeks for 4 cycles with or without low-dose rucaparib (RUCA) 24–30 mg IV D1–3, every 3 weeks for 4 cycles, followed by weekly low-dose RUCA (either 30 mg IV or 100 mg per os) for 24 weeks. Cisplatin dose was frequently delayed or reduced and 73% of patients completed the 4 cycles of chemotherapy. Weekly RUCA during 24 weeks was completed by only 51% of patients. The 2-year disease-free (DFS) was not significantly different between the arms, but an advantage was suggested for RUCA in patients treated with an anthracycline-based preoperative regimen [29], whereas the mutational status had no impact. The low-dose schedule used in this study might have limited the possible impact of RUCA.

The OlympiA study (NCT02032823) was a multicentric, randomized, double-blind, placebo-controlled phase III trial enrolling gBRCA1/2m high-risk HER2-negative BC, after completion of local treatment and (neo)adjuvant chemotherapy (Figure 3). Initially designed to enroll only TNBC patients, the protocol was amended to also include HER2-negative, HR+ BC, provided they had high-risk features. High-risk was defined as follows:For patients with initial surgery: TNBC with macroscopic node-positive disease (>pN1) or pN0 with tumor size > 2 cm; or hormone receptor-positive disease with at least 4 positive lymph nodes (>pN2);For patients receiving neoadjuvant chemotherapy: Invasive residual disease (breast or lymph node) for any TNBC and for hormone-receptor positive disease with high-risk features according to a combined staging system including pre-treatment clinical stage, final pathological stage (CPS), as well as ER status and nuclear grade (CPS + EG).

Patients were randomized between OLA 300 mg (tablets) or placebo BID during 12 months. Patients were stratified according to hormone receptor status, adjuvant or neoadjuvant chemotherapy, and platinum pre-exposure. The primary endpoint is invasive DFS and secondary endpoints include OS, distant metastasis-free survival, incidence of new cancers (including new primary contralateral breast cancers, and new primary ovarian cancer, fallopian tube cancer, and peritoneal cancer), patient-reported outcomes, safety, and exploratory analyses aiming to identify predictive or prognostic factors, as well as potential mechanisms of resistance to treatment. Approximately 1500 patients were randomized and recruitment was closed in 2019. A first interim analysis is expected before the end of 2020. Of note, most patients in OlympiA trials are to receive adjuvant OLA due to residual disease after neoadjuvant chemotherapy. Even though this setting allows selecting patients with a high-risk of recurrence, it might also enrich the population with chemo-resistant disease, a biological subset which might not be the most sensitive to PARP inhibition.

Other clinical trials investigating OLA in the adjuvant setting include the SUBITO (Substantially Improving the Cure Rate of High-risk BRCA1-like Breast Cancer trial; NCT02810743) sponsored by NKI, Netherlands. This study is enrolling patients with stage III breast cancer treated with adjuvant or neoadjuvant chemotherapy and with features of HRD, as defined as either known gBRCA1/2m or BRCA1-like copy number profile evaluated on tumor tissue. The control arm includes dose-dense AC, followed by carboplatin–paclitaxel, before or after local treatment and 1-year adjuvant OLA will be added after completing the standard therapeutic procedures, based on the assumption of positive results for the OlympiA trial. The experimental arm substitutes 2 courses of intensified alkylating chemotherapy (high-dose of carboplatin, cyclophosphamide, and thiotepa) with autologous peripheral stem cell transplantation for the carboplatin–paclitaxel and the OLA sequence. The research hypothesis is that in high-risk BRCA1-like BC, intensified chemotherapy using high-dose alkylating agents will outperform a sequential combination of conventionally-dosed platinum-based chemotherapy and PARPi.

## 5. Other Approaches Under Investigation

Several studies are investigating short pre-surgical exposure to PARPi through the so-called “window-of-opportunity (WOO)” trials, in which the baseline tumor biopsy was obtained and compared to surgically-removed post-treatment tumor tissue. The primary objective was to describe drug-induced molecular changes in the tumor and correlate them with therapeutic responses in order to get insight into the mechanisms of action and resistance. Ongoing WOO trials are currently evaluating OLA alone (NCT04041128, NCT03740893) or in combination with anti-PD-L1 durvalumab (NCT03594396).

Another appealing strategy in the early setting might be a combination of PARPi with radiation therapy [30]. As illustrated in preclinical studies, including breast cancer cell lines or xenograft [31,32,33], PARPi might be a powerful radiosensitizer, with unrepaired SSBs leading to a replication fork collapse, and an increase in DSBs. Of note, a synergistic effect between ionizing radiation (IR) and PARPi might be obtained at doses lower than those able to induce cytotoxicity as a single-agent. Ongoing trials combining IR and OLA in breast cancer include a phase I investigating the combination of OLA with radiation therapy in locally advanced, inflammatory, or metastatic TNBC, as well as TNBC with residual disease surgery (NCT03109080), a phase I study (NCT03598257) examining the same association in inoperable BC (either primary or after local recurrence), including inflammatory BC (IBC), and a randomized phase II trial evaluating the association of OLA + radiotherapy vs. radiotherapy alone in IBC, after neoadjuvant chemotherapy and mastectomy (NCT03598257). Safety and tolerance of VELI was also evaluated in combination with radiation therapy in patients with node-positive disease receiving neoadjuvant chemotherapy and with residual disease (NCT01618357), as well as in loco-regionally recurrent BC or IBC (NCT01477489). Niraparib (NCT03945721) and rucaparib (NCT03542175) were also tested in association with radiation therapy in TNBC with residual disease after neoadjuvant chemotherapy.

Finally, data from a Japanese randomized phase III trial indicated that post-operative capecitabine might increase the overall survival in TNBC with residual disease, after neoadjuvant chemotherapy [34], which has popularized such a strategy in this subtype. Similarly, the association of pembrolizumab to neoadjuvant chemotherapy and its continuation after surgery significantly improved pCR and might improve event-free survival [35]. Based on the hypothesis of higher tumor neoantigen burden in BRCAness tumors, the PARPi-immunotherapy combination is being tested. Recent data from early phase trials in advanced BC patients (TNBC, gBRCA1/2m, or HRD-positive) indicate potential benefits when associating immune checkpoint inhibitor and PARPi [36,37]. Since TNBC are significantly enriched in gBRCA1/2m, combining PARPi and any of these adjuvant therapeutics might warrant examination in the early setting. Thus, Pustzai et al. have reported the results of a sub-cohort of I-SPY2 study, at the AACR 2020 meeting, in which 73 HER2-negative BC patients candidate to pre-operative chemotherapy received durvalumab (an anti-PD-L1 monoclonal antibody) 1500 mg IV Q4W for 3 cycles, with low-dose olaparib 100 mg PO BID from weeks 1 to 11, in combination with weekly paclitaxel (80 mg/m²) × 12, followed by AC × 4, as neoadjuvant chemotherapy. Patients had either TNBC or ER+ tumors with a high-risk Mammaprint signature, but no data was available regarding the BRCA status. The pCR rate was 37% vs. 20% in the shared ISPY-2 control arm, including 47% vs. 27% in TNBC and 28% vs. 14% in ER + BC. There were more toxicity in the experimental arm (febrile neutropenia and immune-related adverse events) but no unexpected event was recorded [38]. Yet, since immunotherapy alone combined with neoadjuvant chemotherapy was demonstrated to improve pCR [35], it was difficult to separately evaluate the relative impact of durvalumab and olaparib in this study.

## 6. Predictive Biomarkers

The identification of potential mechanisms of resistance or predictive factors for response is a critical issue to envision a more rationalized use of this promising class of compounds in early BC patients. One of the best described mechanisms of resistance to PARPi includes the acquisition of secondary somatic BRCA mutations, restoring an open reading frame (reversion mutation) and thereby a functional BRCA protein, allowing an optimal DNA repair by HR and preventing PARPi-induced cytotoxicity. This phenomenon was extensively described in clinical samples that have become resistant to platinum-based or PARPi treatments, in various tumor models, recently including breast cancer [39,40,41]. Yet, such alterations are essentially mechanisms of acquired resistance and little is known about the pathways involved in the primary failure of PARPi. Indeed, results from clinical trials in advanced BC patients with gBRCA1/2m indicate that at least 40% of patients have no clinical benefit when receiving PARPi as a single-agent, demonstrating that, even in such a highly selected subgroup of patients, primary resistance does occur. A better understanding of the biological basis of this phenomenon should help to maximize benefits in the early setting. The absence of BRCA locus-specific loss of heterozygosity in germline BRCA1 and BRCA2 carriers is another potential mechanism of primary resistance [42], even if the topic is debated and evidences on breast cancer are still lacking. Other alterations potentially associated with PARPi resistance might also involve non-BRCA molecules implicated in HR-based DNA repair or fork protection systems [43,44]. 

Another important issue is the potential use of PARPi beyond the gBRCA1/2m carriers, in the so-called “BRCAness” population. Tumors belonging to this phenotype lack gBRCA1/2m but still have major perturbations in HR-based DNA repair, and thus, should have a similar behavior in terms of response to DNA-damaging agents, such as PARPi. The “BRCAness” phenotype might result from either hypermethylation of gene promoter or somatic mutations of BRCA genes. It might also be related to molecular alterations in other genes that are involved in the HR process. Such alterations might result in structural rearrangements of the genome, such as loss of heterozygosity, telomeric allelic imbalance, and large-scale transitions, a genomic scar that might be identified by array CGH or next-generation sequencing technologies, including commercially available signatures [45]. In addition, tumor tests with functional read-out that predict whether HR-based DNA repair actually works independently of the molecular make-up of the tumor might also be useful. As an example, the immunohistochemistry-based detection of RAD51 foci on tumor tissue, a robust sign of active DNA repair only present in the case of intact HR, was recently associated with clinical resistance to PARPi [41,46]. Yet, even though “BRCAness” tumors were shown to have sensitivity to DNA-damaging drugs and to benefit from PARPi in advanced ovarian cancer [47,48,49], whether the same holds true in BC patients, remains to be established. In this latter tumor type, only patients with germline BRCA mutations were demonstrated to benefit from PARPi to date.

## 7. Toxicity Issues in the Early Setting 

A critical question before a possible incorporation of PARPi in the early setting is the tolerance profile. Even though all PARPi share a similar mechanism of action, there are some substantial differences in terms of side effects, which might rely upon off-target activities, different specific inhibitory activities on individual PARP family members (which are known to play distinct and specific biological roles) [50], or different PARP-trapping activity, the latter being related to the potential of myelosuppression [51]. As an example, rucaparib is the only PARPi able to inhibit PARP3, in addition to PARP1 and PARP2, and PARP3 was shown to activate PARP1, making it possible that rucaparib might result in a more potent PARP inhibition [52]. Regarding the synthetic lethality concept, it was hypothesized that most of the cytotoxic effects should be obtained in the cellular context of HR deficiency, a state theoretically only observed in the tumors cells in which the gBRCA1/2m status was associated with additional somatic events, leading to a loss of function of the BRCA proteins. Yet, it has also been suggested that gBRCA1/2m carriers might be associated with more toxicity after DNA-damaging therapy, because these patients could have subtle perturbations in DNA repair even in healthy cells (haploinsufficiency phenotype). No data are available for a differential sensitivity to PARPi, according to the gBRCA1/2 status (because most of the patients are actually gBRCA1/2m), but retrospective data from ovarian cancer patients receiving platinum-based treatment, do not support such a hypothesis [53].

Main toxicities of PARPi are hematological, most notably anemia. In the two randomized phase III trials reported in BC, which included pre-treated patients with advanced disease (OLA or TALA vs. chemotherapy at physician’s choice), anemia of any grade was present in 40% to 52%, including 16% to 38% grade > 3, and blood transfusion was required in 18% to 38% of patients [14,54]. Other hematological toxicities were common and included neutropenia (27% to 34%) and thrombocytopenia (<10% to 26.9%), but grade > 3 toxicities were less frequent and rarely clinically complicated. Non-hematological toxicity was dominated by gastrointestinal events, notably nausea, which was present in 48% to 52% of patients, including 13% to 14% with grade 2, and vomiting, was noted in 24% to 32%, and with grade 2–3 in 8% to 9%. Fatigue was also reported in 29% to 62% of cases. Other common and usually mild side effects might include constipation, diarrhea, decrease appetite, alopecia, and increase in liver enzymes or creatinine, as well as hypercholesterolemia. Most of above-described toxicities were observed within the first three months of treatment and tended to decrease thereafter. Some other toxicities were rarely reported with PARPi, but needs to be investigated, such as neurological (headache, insomnia), respiratory (dyspnea, cough, nasopharyngitis, pneumonitis), musculoskeletal (back pain, arthralgia), cutaneous (pruritus, photosensitivity, allergic urticarial), and cardiovascular (hypertension and tachycardia with rucaparib) symptoms [50]. Of note, PARPi might cause teratogenicity and embryo–fetal toxicity, but no clear impact has been documented on menstruations when talazoparib was administered during six months in the neoadjuvant setting, which might be of interest in a patient population of younger women, with a potential pregnancy project.

Regarding the PARPi mechanistic effects on DNA repair, concerns have emerged on possible induction of secondary cancers, most notably malignant hemopathies. In ovarian cancer patients, in whom the experience of PARPi use is the highest with the highest follow-up, the rate of reported myelodysplastic syndromes or acute myeloid leukemias varies from 0 to 2%, all of these patients were also exposed to platinum-based chemotherapy, making it difficult to definitively conclude on the specific role of PARPi treatment [47,48,55]. Importantly and reassuringly, in OlympiAD, trial no malignant hemopathy was reported to date, while in EMBRACA study there was one AML case in both arms. 

## 8. Conclusions

PARPi have recently demonstrated antitumor activity as a single-agent in advanced BC patients with gBRCA1/2m. When compared with non-platinum chemotherapy at physician’s choice, olaparib and talazoparib significantly improved response rate and PFS, as well as quality of life. Similarly, in combination with platinum-based chemotherapy, veliparib increased PFS. Yet, no OS advantage has been reported to date with any of these compounds in the setting of advanced disease. Only recently PARPi were evaluated in the early setting with nuanced results, while veliparib did not improve pCR when added to neoadjuvant carboplatin-based regimen in unselected TNBC, talazoparib produced impressive results when administered as a single-agent. OlympiA adjuvant trial was recently completed and is currently under analysis. It should allow a precise evaluation of the benefit/risk ratio of 1-year olaparib in gBRCAm having completed the standard treatment for HER2-negative early BC. In addition, the future role of PARPi in the early setting would have to take into account other achievements in the field, such as the use of adjuvant dose-dense chemotherapy [56] or post-neoadjuvant capecitabine [34], which have both been shown to improve OS, notably in TNBC. In this latter subgroup, the potential incorporation of immune checkpoint inhibitors in the neoadjuvant setting, as well as in the post-operative setting, as illustrated in the KEYNOTE-522 study [35], might also question the place of the PARPi in the early setting. Similarly, the increasing use of platinum-based neoadjuvant chemotherapy might alter the sensitivity to adjuvant PARPi when used in the post-operative setting and potentially limit their ultimate impact. In addition, the financial toxicity associated with PARPi, compared to platinum derivatives, with lower costs and already validated efficacy in TNBC, might be challenging. However, the recently reported INFORM study, in which BRCA1/2-germline mutation carriers with HER2-negative early BC were randomized to receive cisplatin 75 mg/m² vs. classical AC, both for 4 cycles before surgery, failed to demonstrate any advantage in terms of pCR for cisplatin in this population [57]. These results suggest that, rather counterintuitively, the platinum-based neoadjuvant regimen might not be the optimal comparator to PARPi, at least in the neoadjuvant setting. Thus, the future of PARPi in an early setting could be an alternative, potentially less toxic but highly effective chemo-free regimen, as illustrated in the TALA neoadjuvant study [27]. A better knowledge of the mechanisms of resistance and accurate predictive biomarkers might help to identify the population most likely to benefit from PARPi. Whether a similar efficacy might be expected in other subgroups of BC patients than the gBRCA1/2m carriers, needs to be determined.

## Figures and Tables

**Figure 1 cancers-12-01378-f001:**
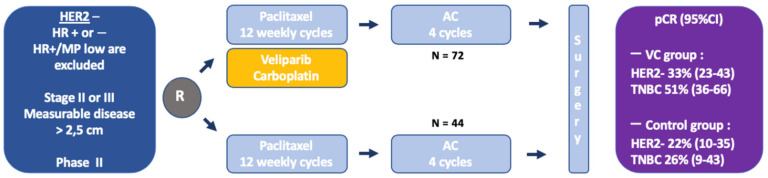
Adaptive Randomization of the Veliparib-Carboplatin Treatment in Breast Cancer (I-SPY2). Weekly paclitaxel 80 mg/m^2^ for 12 doses ± veliparib 50 mg twice a day (BID), and carboplatin AUC 6 on weeks 1, 4, 7, and 10, followed by doxorubicin 60 mg/m^2^ and cyclophosphamide 600 mg/m^2^ (AC) IV, every 2 to 3 weeks for 4 doses. HER2—human epidermal growth factor receptor 2; TNBC—triple negative breast cancer; HR—hormone receptor; pCR—pathological complete response; AUC—area under the curve; RCB—residual cancer burden; MP—MammaPrint; VC—Veliparib/Carboplatin; and AC—doxorubicin and cyclophosphamide.

**Figure 2 cancers-12-01378-f002:**
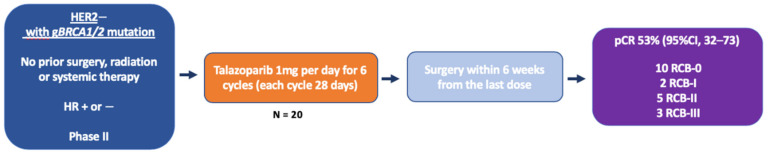
Neoadjuvant Talazoparib for Patients with Operable Breast Cancer With a Germline BRCA Pathogenic Variant. HER2—human epidermal growth factor receptor 2; HR—hormone receptor; pCR—pathological complete response; gBRCA mutation—germline breast cancer 1 or breast cancer 2 mutation; and RCB—residual cancer burden.

**Figure 3 cancers-12-01378-f003:**
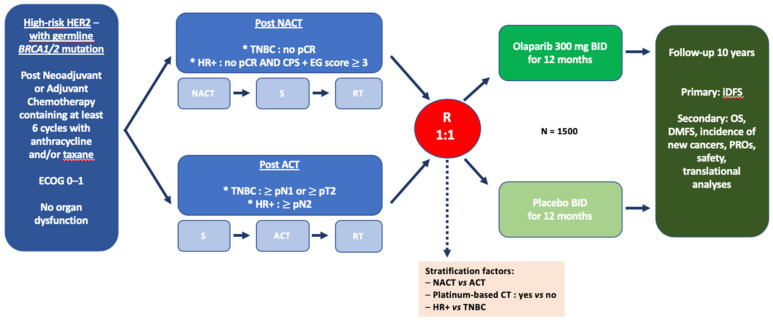
Design of the OlympiA adjuvant study. HER2—human epidermal growth factor receptor 2; TNBC—triple negative breast cancer; ECOG—Eastern Cooperative Oncology Group; HR—hormone receptor; germline BRCA1/2 mutation—germline breast cancer 1 or breast cancer 2 mutation; pCR—pathological complete response; NACT—neoadjuvant chemotherapy; ACT—adjuvant chemotherapy; S—surgery; RT—radiotherapy; iDFS—invasive disease-free survival; OS—overall survival; DMFS—distant metastasis-free survival; and PROS—patient-reported outcomes.

**Table 1 cancers-12-01378-t001:** Main characteristics of major PARPi.

Drug	Pharmaceutical Company	Chemical Formula	2D Structure	Catalytic Site	EMA Registration
Olaparib/AZD-2281	AstraZeneca	C24H23FN4O3	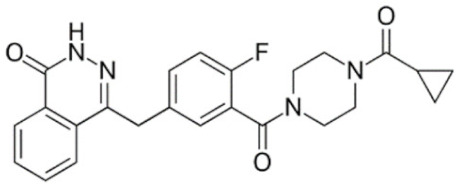	PARP 1, 2 and 3	December 2014
Niraparib/MK-4827	Tesaro/GSK	C19H20N4O	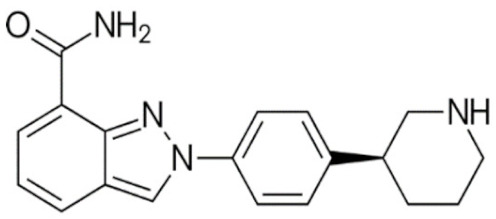	PARP 1 and 2	November 2017
Rucaparib/AG-014699	Clovis oncology	C19H18FN3O	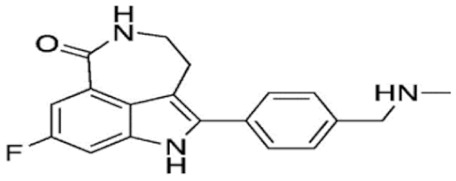	PARP 1, 2 and 3	May 2018
Talazoparib/BMN-673	Pfizer	C19H14F2N6O	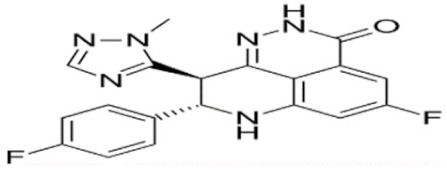	PARP 1 and 2 Powerful PARP trapping	June 2019
Veliparib/ABT-888	AbbVie	C13H16N4O	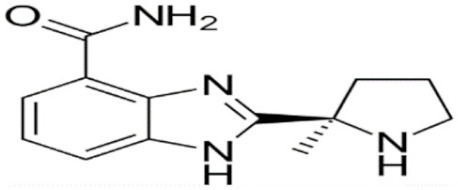	PARP 1 and 2 Weakest PARP trapping	Phase III
Pamiparib/AG-14361	BeiGene	C19H20N4O	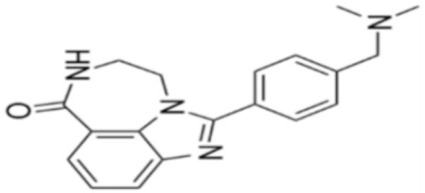	PARP 1	Phase II

PARPi—PARP inhibitors; EMA—European medicines agency; PARP—poly ADP ribose polymerase.

**Table 2 cancers-12-01378-t002:** Completed and ongoing clinical trials testing PARPi in the neoadjuvant setting in early breast cancer.

Study Name/Reference	Daily Dose	Study Phase	Patient Population	Treatment	Primary Outcome/Results
**Veliparib**					
I-SPY 2 [19]	50 mg BID	II Bayesian	TNBC, stage II or III *n* = 39 in veliparib group *n* = 21 in placebo group	Veliparib-C (AUC6, q3 weeks) and standard NACT vs. standard NACT alone	pCR = 51% in exp arm vs. 26% in control
BrighTNEss [22]	50 mg BID	III	TNBC, stage II or III *n* = 316: veliparib group *n* = 160: PC group *n* = 150: P group	Weekly P 12 doses +/− C AUC6 (q3 weeks, 4 cycles) +/− veliparib All patients received AC x4 prior to surgery	pCR = 53% in veli/carbo group vs. 58% in carbo group vs. 31% in paclitaxel alone group
**Olaparib**					
GeparOLA [24]	100 mg BID	II	HER2 negative, HRD Stage I to III *n* = 29 HR+ *n* = 77 HR−	Weekly P + olaparib or C AUC2 for 12 weeks followed by AC	pCR = 55.1% olaparib group vs. 48.6% carbo group
PARTNER NCT03150576	150 mg BID 12 days (D3–14)	II/III	TNBC and/or gBRCAm Stage I to III CP group CP + Olaparib D − 2 group CP + Olaparib D + 3 group	C AUC5 q3 weeks + weekly P +/− olaparib × 4 cycles, followed by anthracycline-based regimen for all before surgery	Feasibility/safety pCR (still ongoing)
**Talazoparib**					
[27]	1 mg	II	HER2 negative, gBRCAm stage I to III *n* = 13 (9 TNBC)	2 months talazoparib monotherapy before standard NACT and surgery	Feasibility/toxicity mDTV = 88% (secondary outcome) RCB 0–I: 10/13
[28]	1 mg	II	HER2 negative, gBRCAm stage I to III *n* = 20	6 months talazoparib	pCR = 53% RCB 0–I = 63%
NCT03499353	1 mg	II	HER2 negative, gBRCAm stage I to III *n* = 60	6 months talazoparib	pCR Ongoing
**Niraparib**					
NCT03329937	200 mg	I	HER2 negative, BRCAm	2 months niraparib followed by surgery directly, or additional niraparib or NACT	Tumor response rate by MRI Safety

TNBC—triple negative breast cancer; BID—twice daily; C—carboplatin; P—paclitaxel; D—day; NACT—neoadjuvant chemotherapy; pCR—pathological complete response; AUC—area under the curve; HRD—homologous recombination deficiency; germline BRCAm—germline breast cancer mutation; RCB—residual cancer burden; AC—anthracycline/cyclophosphamide; mDTV—median decrease in tumor volume; PARPi—poly ADP-ribose polymerase inhibitors; MRI—magnetic resonance imaging; and HR—hormone receptor.

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
