# Peer review of "PARP Inhibitors in the Treatment of Early Breast Cancer: The Step Beyond?"

_cancers, 2020, doi:10.3390/cancers12061378_

Round 1

Reviewer 1 Report

In manuscript ID cancers-802882, entitled "PARP inhibitors in the treatment of early breast cancer: the step beyond?" Authors revised completed and ongoing clinical trials testing PARP inhibitors both in neoadjuvant and adjuvant settings. A discussion of possible interaction of PARP inhibitors with other therapeutic approaches is included,as well as toxicity issues of PARPi were mentioned. In general this is a good work,complete and clear enough. I only suggest to ameliorate the introduction section, inserting a figure or a table reporting the PARPi in clinical development (name and chemical structure).

Author Response

Reviewer #1’s Comments and Authors’ Responses

In manuscript ID cancers-802882, entitled "PARP inhibitors in the treatment of early breast cancer: the step beyond?", authors revised completed and ongoing clinical trials testing PARP inhibitors both in neoadjuvant and adjuvant settings. A discussion of possible interaction of PARP inhibitors with other therapeutic approaches is included, as well as toxicity issues of PARPi were mentioned. In general, this is a good work, complete and clear enough.

We thank the reviewer for his/her positive comment.

I only suggest to ameliorate the introduction section, inserting a figure or a table reporting the PARPis in clinical development (name and chemical structure).

We have added a Table (Table 1) in the Introduction section, which show the name and chemical structure of major PARPis in clinical development. On line 47 (previous pagination PP), we have added the following sentence: “Table 1 shows the name and chemical structure of major PARPi in advanced clinical development.

Reviewer 2 Report

This review titled"PARP inhibitors in the treatment of early breast cancer: the step beyond?" is really interesting. However the author focused only few studies in detail. I feel the author have to provide few more references for the each section.

I have a minor comments to the author:

  1. There are few typo errors should changed.
  2. In the section 2 and 3 have only few references provided. It will be better to provide few more citation. And these section should have overview figure.
  3. The author can include recent finding in conventional and advanced therapies and its adverse events.
  4. The manuscript must be edited by native English speakers.

Author Response

Reviewer #2’s Comments and Authors’ Responses

This review titled "PARP inhibitors in the treatment of early breast cancer: the step beyond?" is really interesting. However the author focused only few studies in detail. I feel the author have to provide few more references for the each section.

We thank the reviewer for his/her positive comment.

I have a minor comments to the author:

1/ There are few typo errors should be changed.

They have been corrected.

2/ The sections 2 and 3 have only few references provided. It will be better to provide few more citations. And these sections should have overview figure.

We thank the reviewer. Indeed, a few references were missing. We have added them in both sections, and the Table 2 (previously Table 1 in the original version) has been amended.

We have added two new figures (Figure 1 in section 2 and Figure 2 in section 3) that show the design and the main results of two major completed trials.

3/ The authors can include recent finding in conventional and advanced therapies and its adverse events.

We thank the reviewer for this pertinent remark, that we understood as a call for discussing more in depth the role of PARPi in early breast cancer patients, in the context of recent findings in conventional and advanced therapies, as we did, partially, in the section 5, starting line 295 (PP) : Finally, data from a Japanese randomized phase III trial indicate that post-operative capecitabine may increase overall survival in TNBC with residual disease after neoadjuvant chemotherapy [29], which has popularized such a strategy in this subtype. Similarly, the association of pembrolizumab to neoadjuvant chemotherapy and its continuation after surgery significantly improves pCR and may improve event-free survival [30]. Based on the hypothesis of higher tumor neoantigen burden in BRCAness tumors, the PARPi-immunotherapy combination is being tested and recent data from early phase trials in advanced BC patients (TNBC, gBRCA1/2m, and/or HRD-positive) indicate potential benefits when associating immune checkpoint inhibitor and PARPi [31,32].”

Accordingly, to address this comment, we have added the following points:

  • Results from the very recently reported I-SPY2 study sub-cohort evaluating the combination of durvalumab and olaparib in association with weekly paclitaxel administered in the neoadjuvant setting. On line 305 (PP), we have added the following sentences:

“Thus, Pustzai et al have reported at the AACR 2020 meeting 2020 the results of a sub-cohort of ISPY-2 study, in which 73 HER2-negative BC patients candidate to pre-operative chemotherapy received durvalumab (an anti-PD-L1 monoclonal antibody) 1500 mg IV Q4W for 3 cycles with low-dose olaparib 100 mg PO BID from weeks 1 to 11, in combination with weekly paclitaxel (80 mg/m²) x 12, followed by AC x 4, as neoadjuvant chemotherapy. Patients had either TNBC or ER+ tumors with high-risk Mammaprint signature, but no data was available regarding to BRCA status. The pCR rate was 37% vs. 20% in the shared ISPY-2 control arm, including 47% vs. 27% in TNBC and 28% vs.14% in ER+ BC. There were more toxicity in the experimental arm (febrile neutropenia and immune-related adverse events) but no unexpected event was recorded [38]. Yet, since immunotherapy alone combined to neoadjuvant chemotherapy was demonstrated to improve pCR [35], it is difficult to evaluate separately the relative impact of durvalumab and olaparib in this study.”

  • A discussion about the integration of PARPi in regard to the already available achievement in early breast cancer patients has been added to the Conclusion section, line 395 (PP): “In addition, the future role of PARPi in the early setting will have to take into account other achievements in the field, such as the use of adjuvant dose-dense chemotherapy [57] or post-neoadjuvant capecitabine [58] which both have been shown to improve OS, notably in TNBC. In this latter subgroup, the potential incorporation of immune checkpoint inhibitors in the neoadjuvant setting, as well as in the post-operative setting, as illustrated in the KEYNOTE-522 study [59], may also question the place of PARPi in the early setting. Similarly, the increasing use of platinum-based neoadjuvant chemotherapy may alter the sensitivity to adjuvant PARPi when used in the post-operative setting and potentially limit their ultimate impact. In addition, the financial toxicity associated with PARPi, compared to platinum derivatives, with lower costs and already validated efficacy in TNBC, may be challenging. However, the recently reported INFORM study, in which BRCA1/2-germline mutation carriers with HER2-negative early BC were randomized to receive cisplatin 75 mg/m² vs. classical AC, both for 4 cycles before surgery, failed to demonstrate any advantage in terms of pCR for cisplatin in this population [60]. These results suggest that, rather counterintuitively, platinum-based neoadjuvant regimen might not be the optimal comparator to PARPi, at least in the neoadjuvant setting. Thus, the future of PARPi in early setting could be as an alternative, potentially less toxic but highly effective chemo-free regimen, as illustrated in the TALA neoadjuvant study [27].”

4/ The manuscript must be edited by native English speakers.

The revised version has been edited by a native English speaker.

Reviewer 3 Report

The manuscript by Gonçcalves et al. covers the innovative and important issue of the use of PARP inhibitors in early breast cancer setting. It exposes with precision and clarity the available data on this setting and also includes two paragraphs on potentially useful biomarkers and toxicities. Although this is not the first review treating PARP inhibitors in early breast cancer (see also doi: 10.1097/CCO.0000000000000516), many novelties here exposed have not been discussed in previous papers. Therefore, I recommend this paper for publication after minor revisions as reported below.

The manuscript is generally well-written in standard English.

I suggest the following minor revisions:

  • Page 2 line 63 and page 9 line 383: please cite the paper describing the OS results for OlympiAD trial (Robson et al, Annals of Oncology 2019, doi: 10.1093/annonc/mdz012) and, if possible, the OS results for EMBRACA trial (discussed at AACR 2020, still not published);
  • Page 2 line 64: the abbreviation of “veliparib” should be made here and not at line 76;
  • Page 2 line 81: “aiming at evaluating”;
  • Page 2 lines 79-104: this part about I-SPY2 trial is quite complicated to read. First of all, I suggest specifying that the trial is still recruiting, thus justifying the use of present tense on describing it. Secondly, Line 89: for the part starting from here describing the benefit of VELI-CARBO a reference is missing and the fact that this is part of the I-SPY 2 TRIAL should be clarified (Rugo et al, NEJM 2016. Doi: 10.1056/NEJMoa1513749);
  • Page 3 line 120: “considered as not eligible for breast conserving surgery AT DIAGNOSIS”;
  • Page 3 line 127: “in BRCAness patients” is incorrect, please correct (e.g. “in patients with homologous DNA repair deficiency”) and add a reference;
  • Page 3 line 194: the results of the NCT03499353 are now published so this part should be updated (doi: 10.1200/JCO.19.01304);
  • Page 6 line 217: please add “cisplatin 75 mg/mq2 D1 every 3 weeks for 4 cycles”;
  • Page 8 line 310-316: I suggest mentioning the locus specific loss of heterozygosity as a mechanism of primary resistance, as discussed in the outstanding paper by Maxwell K. et al in Nature Communications 2017 (doi: 10.1038/s41467-017-00388-9), even if the topic is debated and evidences on breast cancer are still lacking;
  • In the “Conclusions” I suggest mentioning the cost of PARP inhibitors as part of the reasoning about their introduction as early breast cancer treatment, since platinum salts have already been proved as effective in this setting with lower costs.

Author Response

Reviewer #3’s Comments and Authors’ Responses

The manuscript by Gonçalves et al. covers the innovative and important issue of the use of PARP inhibitors in early breast cancer setting. It exposes with precision and clarity the available data on this setting and also includes two paragraphs on potentially useful biomarkers and toxicities. Although this is not the first review treating PARP inhibitors in early breast cancer (see also doi: 10.1097/CCO.0000000000000516), many novelties here exposed have not been discussed in previous papers. Therefore, I recommend this paper for publication after minor revisions as reported below.

We thank the reviewer for his/her positive comment.

The manuscript is generally well-written in standard English.

I suggest the following minor revisions:

1/ Page 2 line 63 and page 9 line 383: please cite the paper describing the OS results for OlympiAD trial (Robson et al, Annals of Oncology 2019, doi: 10.1093/annonc/mdz012) and, if possible, the OS results for EMBRACA trial (discussed at AACR 2020, still not published).

On line 63 (PP), the sentence “Of note, no overall survival (OS) benefit was reported in none of these studies.” has been replaced by the following sentence: “Of note, no overall survival (OS) benefit was reported in OlympiAD trial [14] and EMBRACA trial as recently reported at AACR 2020 meeting.

2/ Page 2 line 64: the abbreviation of “veliparib” should be made here and not at line 76.

We have corrected.

3/ Page 2 line 81: “aiming at evaluating”.

We have replaced “aiming to evaluate” by “aiming at evaluating”.

4/ Page 2 lines 79-104: this part about I-SPY2 trial is quite complicated to read. First of all, I suggest specifying that the trial is still recruiting, thus justifying the use of present tense on describing it. Secondly, Line 89: for the part starting from here describing the benefit of VELI-CARBO a reference is missing and the fact that this is part of the I-SPY 2 TRIAL should be clarified (Rugo et al, NEJM 2016. Doi: 10.1056/NEJMoa1513749);

We agree. On line 81 (PP), we have replaced “an adaptive randomized phase II…” by “an ongoingadaptive randomized phase II…”.

On line 79 (PP), the sentence “It was the first PARPi assessed in the neoadjuvant setting.” has been replaced by the following sentence “It was the first PARPi assessed in the neoadjuvant setting in the carboplatin-VELI arm of the I-SPY 2 trial”.

On line 90 (PP), the sentence “In a study involving 72 patients…” has been replaced by the following sentence “In this trial, the carboplatin-VELI arm enrolled 72 patients…”.

On line 93 (PP), the missing reference has been added.

5/ Page 3 line 120: “considered as not eligible for breast conserving surgery AT DIAGNOSIS”;

On line 122 (PP), the sentence “considered on diagnosis as not eligible for breast conserving surgery …” has been replaced by the following sentence “considered as not eligible for breast conserving surgery at diagnosis…”.

6/ Page 3 line 127: “in BRCAness patients” is incorrect, please correct (e.g. “in patients with homologous DNA repair deficiency”) and add a reference;

On line 129 (PP), the sentence “in BRCAness HER2-negative patients” has been replaced by the following sentence “in HER2-negative patients with homologous DNA repair deficiency”. The missing reference has been added.

7/ Page 3 line 194: the results of the NCT03499353 are now published so this part should be updated (doi: 10.1200/JCO.19.01304);

This reference (doi: 10.1200/JCO.19.01304) does not correspond to the ongoing NCT03499353, but corresponds in fact to the update of our initial reference 23: we have now updated this reference in the text and in the new Table 2.

8/ Page 6 line 217: please add “cisplatin 75 mg/mq2 D1 every 3 weeks for 4 cycles”;

On line 220 (PP), the sentence “cisplatin 75 mg/m² on D1 for 4 cycles or cisplatin 75 mg/m² on D1 combined with low-dose rucaparib (RUCA) 24-30 mg IV D1-3 for 4 cycles” has been replaced by the following sentence “cisplatin 75 mg/m² D1 every 3 weeks for 4 cycles without or with low-dose rucaparib (RUCA) 24-30 mg IV D1-3 every 3 weeks for 4 cycles.”

9/ Page 8 line 310-316: I suggest mentioning the locus specific loss of heterozygosity as a mechanism of primary resistance, as discussed in the outstanding paper by Maxwell K. et al in Nature Communications 2017 (doi: 10.1038/s41467-017-00388-9), even if the topic is debated and evidences on breast cancer are still lacking;

On line 314 (PP), we have added the following sentence: “The absence of BRCA locus specific loss of heterozygosity in germline BRCA1 and BRCA2 carriers is another potential mechanism of primary resistance [42], even if the topic is debated and evidences on breast cancer are still lacking.

10/ In the “Conclusions” I suggest mentioning the cost of PARP inhibitors as part of the reasoning about their introduction as early breast cancer treatment, since platinum salts have already been proved as effective in this setting with lower costs.

As already mentioned in our response to Reviewer #2, we have taken into account this comment by adding the following sentence at the end of the Conclusion: “Similarly, the increasing use of platinum-based neoadjuvant chemotherapy may alter the sensitivity to adjuvant PARPi when used in the post-operative setting and potentially limit their ultimate impact. In addition, the financial toxicity associated with PARPi, compared to platinum derivatives, with lower costs and already validated efficacy in TNBC, may be challenging.”
